# Barriers and facilitators to healthy lifestyle and acceptability of a dietary and physical activity intervention among African Caribbean prostate cancer survivors in the UK: a qualitative study

Vanessa Er,[1] J Athene Lane,[1,2] Richard M Martin,[1,2,3] Raj Persad,[4] Frank Chinegwundoh,[5,6] Victoria Njoku,[5] Eileen Sutton[1]

For numbered affiliations see end of article.

**Correspondence to**
Dr Vanessa Er;
vanessa.er@bristol.ac.uk

## ABSTRACT

**Objectives** Diet and lifestyle may have a role in delaying prostate cancer progression, but little is known about the health behaviours of Black British prostate cancer survivors despite this group having a higher prostate cancer mortality rate than their White counterparts. We explored the barriers and facilitators to dietary and lifestyle changes and the acceptability of a diet and physical activity intervention in African Caribbean prostate cancer survivors.

**Design** We conducted semistructured in-depth interviews and used thematic analysis to code and group the data.

**Participants and setting** We recruited 14 African Caribbean prostate cancer survivors via letter or at oncology follow-up appointments using purposive and convenience sampling.

**Results** A prostate cancer diagnosis did not trigger dietary and lifestyle changes in most men. This lack of change was underpinned by five themes: precancer diet and lifestyle, evidence, coping with prostate cancer, ageing, and autonomy. Men perceived their diet and lifestyle to be healthy and were uncertain about the therapeutic benefits of these factors on prostate cancer recurrence. They considered a lifestyle intervention as unnecessary because their prostate-specific antigen (PSA) level was kept under control by the treatments they had received. They believed dietary and lifestyle changes should be self-initiated and motivated, but were willing to make additional changes if they were perceived to be beneficial to health. Nonetheless, some men cited advice from health professionals and social support in coping with prostate cancer as facilitators to positive dietary and lifestyle changes. A prostate cancer diagnosis and ageing also heightened men's awareness of their health, particularly in regards to their body weight.

**Conclusions** A dietary and physical activity intervention framed as helping men to regain fitness and aid post-treatment recovery aimed at men with elevated PSA may be appealing and acceptable to African Caribbean prostate cancer survivors.

### Strengths and limitations of this study

► Our study is one of the first to explore the dietary and lifestyle practices and information seeking behaviour of African Caribbean men after a prostate cancer diagnosis in the UK.

► Views and preferences for a dietary and physical activity intervention were sought from men from diverse sociodemographic backgrounds and with different prostate cancer clinical history.

► We struggled to recruit men on active monitoring. It has been reported that Black African and Caribbean men in the UK were more likely to undergo radical treatment compared with their White counterparts, perhaps due to a tendency for their prostate cancers to be managed more aggressively by clinicians.

► We did not interview men's partners but it would be of interest to explore eating habits and food choices of the family with their partners to gain a deeper understanding of the roles and responsibilities regarding food within the household.

► This study reflects only intentions in response to a dietary and physical activity intervention which may differ from that in a real intervention.

## BACKGROUND

Prostate cancer is the second most common cancer in men worldwide, with over 1.1 million cases diagnosed in 2012.[1] In the UK, there are over 40 000 newly diagnosed prostate cancer cases and 10 000 deaths each year.[2] It affects Black men (Black African, Black Caribbean and Other Black) disproportionately, and in England, they are at twice the risk of being diagnosed with and dying from prostate cancer than White men of the same age.[3] The African Caribbean community in the UK is established and make up a large proportion of older adults

within the Black population;[4] hence, they are more likely to be affected by prostate cancer as it is strongly related to age. Awareness of prostate cancer among Black men is low, with widespread misconceptions about the methods of diagnosis and treatment.[5] Their reluctance to be tested for prostate cancer has been attributed to a lack of communication and information from health professionals, mistrust of the healthcare system, perceived threats to masculinity, and fear of cancer and side effects of prostate cancer treatment.[5]

Radical treatments for prostate cancer are often accompanied by adverse effects that negatively impact on men's quality of life,[6] and the prostate-specific antigen (PSA) test cannot distinguish the majority of men with indolent cancer from the minority with aggressive, fatal cancer.[7] Since diet and lifestyle are thought to play a pivotal role in prostate carcinogenesis,[8] attention has turned towards the development of dietary and lifestyle interventions to prevent the progression of prostate cancer. A dietary and lifestyle intervention may appeal to first generation African Caribbean prostate cancer survivors in the UK if it is congruent with their health beliefs -, as they have been found to have a preference for natural treatment and place great emphasis on food for health maintenance.[9 10] These beliefs could be linked to their memories of growing up in the Caribbean, where fresh and natural foods are in abundance, and the use of herbal remedies is widespread.[9 10]

In 2007, the World Cancer Research Fund and the American Institute for Cancer Research (WCRF/AICR), which published the most comprehensive and authoritative report on environmental exposures and cancer, concluded that there is some evidence that selenium and foods rich in lycopene probably decrease the risk of prostate cancer while diets high in calcium increases prostate cancer risk.[8] This was followed by the WCRF/AICR continuous update project report in 2014, which found strong evidence for a link between body fatness and increased risk of advanced prostate cancer.[11] Physical activity has been linked to a lower risk[12 13] of prostate cancer progression and mortality and has also been shown to alleviate the adverse effects of prostate cancer treatment.[14] However, current evidence suggests that prostate cancer survivors have suboptimal diet and lifestyle profiles[15–18] and the evidence-base to support the development of successful and conclusive interventions is lacking and of poor quality.[19] There is a high prevalence of obesity among prostate cancer survivors,[15 17 18] and the proportion of prostate cancer survivors consuming '5 a day' of fruits and vegetables in most studies was relatively low, ranging from 15.6% to 30.7%.[15 16 18] Similarly, it has been reported that over 50% of prostate cancer survivors are inactive.[15–17] While there is a body of literature on the health behaviours of prostate cancer survivors, these studies were conducted in predominantly White population and in USA, with limited qualitative studies exploring the facilitators of and barriers to dietary and lifestyle changes.[20–24]

Dietary studies of African Caribbean in the UK (mainly first-generation) found that they have a lower energy and fat intake and higher fruit and vegetable intake compared with the British population,[25] as they tend to adhere to a traditional African Caribbean diet that is rich in complex carbohydrates, including pulses, fruits and vegetables. The impact of dietary acculturation on second generation African Caribbean in the UK was evident, with this group more likely to adopt a Westernised diet which is high in fat and sugar and low in fruits and vegetables.[26] To our knowledge, there is no qualitative study on the health behaviours of African Caribbean men after a prostate cancer diagnosis. It is important to understand the health behaviours and needs of African Caribbean prostate cancer survivors, so as to inform the development of a dietary and lifestyle intervention which is both acceptable to this population and effective in improving prostate cancer outcomes. The aim of this qualitative study was to explore the facilitators and barriers to dietary and lifestyle changes and the acceptability of a dietary and physical activity intervention among African Caribbean prostate cancer survivors to inform the development of a high-quality diet and physical activity randomised controlled trial.

## METHODS
### Study population
We recruited men aged 18 and above who self-identified as African Caribbean and had a clinically confirmed prostate cancer diagnosis from two large cities in the UK by invitation letter through their consultants and in-person at oncology follow-up clinics. We used purposive sampling to select men across a range of ages, socioeconomic positions, marital status and treatment types to obtain a representative sample.[27] These factors have been identified in the literature to influence dietary and lifestyle changes after a prostate cancer diagnosis.[28] As a result of slow recruitment at the first site and time constraints—this study was conducted as part of a doctoral degree—we used convenience sampling[27] and focused our recruitment on another site with a larger African Caribbean community. We aimed to recruit between 15 and 20 participants. The sample size was expected to obtain sufficient data to detect possible themes based on previous research[28 29] and advice from experienced qualitative researchers.[30–32] We were unable to recruit men on active monitoring. Nonetheless, no new themes were emerging from the data after 14 participants had been interviewed.

### In-depth interviews
All participants provided their written informed consent to take part in the study, for their interviews to be audio-recorded and quotes to be published. A researcher (VE) carried out in-depth, semistructured interviews with participants at their homes between 2014 and 2015. Interviews lasted between 45 min and 1 hour 40 min and participants received a £15 supermarket voucher at the end of

the interview as a thank you for taking part. VE presented herself as a student who would like to know the participants' view on diet and prostate cancer as part of her studies. Although she has a Masters degree in nutrition, she did not reveal this to the participants. Her interest in researching diet and lifestyle of people from minority ethnic groups is due to her experience of growing up in Malaysia, a multiracial and multicultural country and was surprised to discover that despite being disproportionately affected, there was a lack of research on the health behaviours of Black men in the UK diagnosed with prostate cancer.

VE used an interview topic guide (see online supplementary file 1) to direct discussions on men's dietary and lifestyle practices before and after a prostate cancer diagnosis, the reasons for dietary and/or lifestyle changes (or lack of), sources and views on dietary and lifestyle information, and views on a proposed dietary and physical activity intervention for prostate cancer survivors, which consists of increasing tomato intake, a dairy-free diet and 30 min of brisk walking a day. The topic guide was informed by the literature, reviewed by a public and patient involvement group, and was refined twice based on preliminary findings from early interviews to include topics on body weight, familiarity with soya products, and tomatoes being a feature in the Caribbean diet.

## Data analysis

The interviews and data analysis were carried out concurrently.[33] This was an iterative process such that salient issues or topics raised in early interviews could be explored or follow-up in subsequent interviews. All interviews were anonymised, transcribed verbatim and analysed using thematic analysis[34] with the aid of a qualitative data analysis computer software package (NVivo V.10). The researcher (VE) used an inductive approach to code the data in 'chunks' to keep the context where the data originated and developed an initial coding frame.[34] A portion of transcripts was coded independently by another researcher (ES) to ensure coding was consistent and was a balanced representation of participants' views. Throughout the coding process, extracts of data across the dataset were compared to check for similarities and differences.[34] The coding framework was refined and revised by VE and ES as the analysis progressed. We identified, grouped and conceptualised patterns within and across the dataset which are salient to the participants into themes.[34] Field notes and analytic memos written by VE were used to explore how codes and themes related to each other and aided the conceptualisation of key themes. VE and ES discussed any discrepancies and finalised the themes. All names presented here refer to pseudonyms.

## RESULTS

The characteristics of the participants are presented in table 1. At the time of interview, the age of the participants ranged from 52 to 80 years, with a median age of 71.5 years. Most of the men were married or living with a partner; diagnosed with prostate cancer less than 5 years at the time of the interview and had radiotherapy and adjuvant hormone therapy as treatment. With the exception of five men, the rest of the participants were retired. Most men have or previously had semiroutine or routine occupation.

Five themes were identified from the analysis which captured the barriers and facilitators to dietary and lifestyle changes, the rationale for seeking (or not) dietary and lifestyle information and men's views on the target population, timing and acceptability of a proposed dietary and physical activity intervention. Additional quotes relating to each theme are provided in online supplementary material 2.

### Theme 1: Pre-cancer diet and lifestyle

Most men who perceived their diet to be healthy before a prostate cancer diagnosis ate as usual. Some men had already made positive dietary changes prompted by other health problems such as diabetes, while others questioned what changes could be made if they were already eating healthily.

'Well, but erm - change to what? (Laugh) What do you change to, in those circumstances? If you eat, you know, if you eat healthy food, meat, vegetables, salads, fruits, what do you change to?' (Jonah)

Thus, most of the men did not seek further dietary information after a prostate cancer diagnosis as they perceived themselves as having a healthy lifestyle and knowing what constitutes a healthy diet. When men were asked for their views on a dietary intervention which would require them to eat more tomatoes and cut out dairy products, they were amenable to it because it complemented their existing diet. They explained that tomatoes are commonly consumed as part of a traditional Caribbean diet, while dairy products only account for a small fraction of it. Similarly, a proposed physical activity intervention which involves brisk walking for 30 min a day was considered achievable, with several men already doing it in their daily lives.

Men who perceived themselves to be active before prostate cancer also did not think they needed to increase their physical activity. For men, being active meant being up on their feet and 'doing things', but they advocated moderation. As one participant (Errol) noted:

'I do the exercise in my work. 'Cause when you're a builder, you don't sit, you don't do nothing, you have to move around all the time, you know. So that to me enough exercise.'

A few men wanted to be more active but they were impeded by health conditions or sports injuries. Conversely, men who made changes acknowledged that their diet and lifestyle before prostate cancer could have been better and a prostate cancer diagnosis acted as a catalyst for the change.

**Table 1** Characteristics of participants

| Pseudonym | Age | Marital status | Years since diagnosis | Cancer stage | Treatment | Comorbidities |
|---|---|---|---|---|---|---|
| Aaron | 79 | M/P | ≤1 | Localised | Not started* | Hypertension, gastro-oesophageal reflux disease |
| Dennis | 56 | M/P | ≤1 | Localised | Not started* | Hypertension |
| Jonah | 75 | M/P | >10 | Locally advanced | RT/HT | – |
| Fabian | 73 | M/P | >1 and <5 | Locally advanced | RT/HT | Type 2 diabetes |
| Colton | 77 | M/P | >10 | Locally advanced | RT/HT | Hyperthyroidism |
| Albert | 65 | M/P | ≥5 and <10 | Localised | RT/HT | Hay fever |
| Jamal | 70 | S/W | >1 and <5 | Locally advanced | RT/HT | – |
| Matthew | 80 | S/W | >10 | Locally advanced | RT/HT | Type 2 diabetes |
| Irvin | 79 | S/W | >1 and <5 | Localised | RT/HT | – |
| Errol | 76 | S/W | ≤1 | Locally advanced | RT/HT | Type 2 diabetes |
| Shaun | 68 | M/P | ≤1 | Localised | RT/HT | Type 2 diabetes, asthma |
| Joseph | 63 | M/P | ≤1 | Localised | Prostatectomy | – |
| Sebastian | 61 | M/P | ≤1 | Localised | Prostatectomy | – |
| Thomas | 52 | M/P | ≤1 | Localised | Prostatectomy | – |

*Had not started treatment at the time of interview.
M/P, married or living with a partner; RT/HT, radiotherapy and adjuvant hormone therapy; S/W, single or widowed.

'…But since I was diagnosed, er, we just, I would say about, the beef probably come out, we cut down beef a bit, you know? … they always say too much red beef is not good for you and that. So we just cut it down, you know?' (Albert)

### Theme 2: Evidence: link between diet, lifestyle and prostate cancer

Men's views on the causes of cancer varied, ranging from genetic factors, fate or chance, stress (not enough rest), diet and lifestyle to food production (chemicals). However, the majority of men were aware that Black men have higher risk of prostate cancer and some described it as an 'old man's complaint'. A few men cited a lack of evidence that prostate cancer is linked to diet and lifestyle to support their view. This was contrasted with men's belief that smoking increases the risk of cancer due to widely established evidence that it causes lung cancer.

'…up until now they haven't come out with err, err, anything positive to say, "This is the result, this is the main reason why black men from the Caribbean or West Africa have a, how I say, the most higher rate."… So I don't think that I am in any position to think of, because of our diet or because of what we eat, do you understand?' (Dennis)

Men who believed that prostate cancer is not connected to diet were less likely to seek dietary information. This was partly due to their mistrust of the dietary messages from media which they regarded as conflicting. They preferred receiving information from health professionals who they regarded as experts and a trusted source of health information, and that had a positive influence on their health behaviour. For example, one man (Colton) started eating tomatoes and drinking pomegranate juice because his clinicians informed him that it could be beneficial for prostate cancer:

'she [dietician] said with, with the prostate cancer that I've got I should eat a lot of tomato. Tomato is good for the prostate cancer. My doctor before, Doctor M, told me that if I drink pomegranate juice it's a little bit helpful as well…That's what I buy, we buy, things, we buy pomegranate juice.'

However, only a few men reported that they received dietary or lifestyle advice from health professionals, with some explaining that their doctor was uncertain about the effectiveness of diet and lifestyle on prostate cancer.

There was however a small number of men who believed that diet and lifestyle are important for cancer prevention, but they did not think diet and lifestyle have any impact on cancer progression, particularly if their treatment was effective:

'So, so what the difference that it make to, it's not going to affect me now because I'm, I've [passed that stage] with the prostate cancer.' (Albert)

Consequently, men suggested that a dietary and physical activity intervention should be offered before treatment, as the effect of the intervention would not be confounded by the treatment and if the intervention works, treatment would not be necessary. The significance of PSA became apparent when men explained their views, as a high or rising PSA level after treatment was perceived as indicative of recurrent prostate cancer.

'Well, do it before treatment 'cause it (PSA level) might move and it might not. 'Cause if it moves then obviously I've got to go in for treatment. And if it doesn't move, obviously the longer it stays away then the better it is for me. And I wouldn't actually need the radiotherapy.' (Sebastian)

Accordingly, they believed that interventions should be targeted at men who have a high PSA level and were more likely to adopt dietary or lifestyle advice if it was shown to reduce or keep PSA level low.

### Theme 3: Coping with prostate cancer: just get on with it

Men had differing approaches to coping with prostate cancer which influenced whether they made positive changes to their diet and lifestyle. Those who were sanguine about prostate cancer tended to 'take it as it comes' and thus less likely to change their diet and lifestyle and were not interested in gaining further dietary and lifestyle information. Similarly, men who considered their cancer as treated just wanted to move on and not dwell on it further.

'Well me don't really want to get no more information, because so far I'm not living with the thing again. So I don't really want to get no more information, I just try to do, the, my best, try to help myself in many ways.' (Jamal)

These approaches were directly contrasted with those of men who were actively seeking ways to alleviate the side effects they experienced and wanting to be fit again. Thomas spoke about exercise as a way of making himself feel better as it felt like he was doing something for his body, rather than relying on medication for depression or pain:

'If there's something that you need, to motivate yourself and exercise, might trigger it off. I'm not sure, but I know that sitting down, taking tablets, isn't going to do it, or not even diet, to that point.'

It became evident that social networks and support were important for men in coping with prostate cancer and facilitated positive dietary and lifestyle changes. Partners, family and friends acted as a 'change agent' to enable these changes. The role of partner or wife in initiating dietary changes and encouraging men to live healthier is highlighted in this comment:

'… more of less it's the wife that is leading the process (dietary change) really. (Laugh) Yeah, she's the one doing the shopping, she's the one providing the food. And now she's, she's going and, and do all the shopping and buy whatever she thinks that's good for our health.' (Dennis)

Several men commented on the importance of social support to stay physically active. They were very positive about group exercise citing shared purpose and motivation as an attraction to them, as one participant (Matthew) commented:

'what group exercises do is to encourage you and encourage others to follow. That is why I used to like going to the gym, 'cause you had people there who were there for the same purpose.'

Whether men made dietary and lifestyle changes was also dependent on their priorities and concerns after cancer treatment. Men typically experienced side effects from treatments including incontinence and erectile dysfunction. Thus, coping and adapting to these side effects were their main priorities. Primarily, incontinence was reported as a barrier to physical activity.

'I want to go to the gym soon, but like I said, because I'm so wet I won't have – I'm hoping that it will ease up a little bit so I can get to do something else…' (Sebastian)

Loss of income because of a prostate cancer diagnosis could also have a knock-on effect on opportunities for physical activity. Sebastian who was self-employed had financial worries and his main priority was to return to work, for without an income, he could not afford to pay for gym membership.

### Theme 4: Ageing: those were the days, this is me now

Most men used to play sports and acknowledged the health benefits of physical activity, but old age was the most commonly cited reason for not doing strenuous physical activity. For men who had retired, life after work was portrayed as a time to rest (slow down) and for leisure. Most men (aged 70 and above) viewed themselves as too old to be playing sports and gentle exercise as safer and more appropriate for their age. Therefore, a brisk walking intervention was perceived as safe and acceptable by men in the study.

'I'm too old to go to the gym man. I don't want to go to the gym to get a heart attack. (Laughter) Because I don't know, I don't go there [to lift up] these big great things, I go…on the treadmill. (Laugh) I used to go on it but I don't want to do all these type of things now.' (Jamal)

Men made frequent references to their body when describing their physical limitations. They also talked about having to accept and adapt to the changes in their body as they aged. As one participant (Joseph) noted:

'Age is against me now (laugh). I don't feel much pain, but you can tell on the body. Your body can't cope with that. I know it can't cope with it anymore.'

Younger men (under 65 years old) however had no qualms about performing higher intensity physical activity. For Thomas, being back in the gym and having the ability to do the exercises just as before, was a sign of recovery:

'erm- because I've been in a gym before, it's going back to an old friend, and actually familiarising yourself with it, and being told- and telling yourself that you- you're back on the road to recovery…To be back in that gym environment again, it- it was lovely…'

Similarly, Dennis who had a prostate biopsy before the interview hoped to continue playing cricket for his club:

'Well, I used to play every Sunday…but since I had the operation I haven't played…when I am feeling healthy and okay I will.'

On the other hand, men had more awareness of their health as they aged because they perceived that they were more susceptible to illness, so having a healthy diet and lifestyle became more relevant to them. For the men in this study, ageing also shifted the focus from providing for the family to taking care of themselves. There was also a perception espoused by men that getting older made one wiser and thus more sensible. Some men noticed that they put on weight more easily as they aged, mainly because they were less active than before. That prompted men to eat less, cut down food portion sizes, alcohol intake or exercise more.

'Yeah, that's why I said don't want to eat fatty things. I don't want to be bloated out, put it that way…Yeah, yeah, I don't want to get big, fat and clumsy man, where I can't move about and things like that. So I, I watch, I know what I'm eating.' (Jamal)

The younger men who experienced side effects from prostate cancer treatments, especially incontinence and fatigue and thus unable to perform strenuous physical activity also expressed concerns about their weight and wanted to regain fitness to cope with these side effects.

'… she [trainer] said, "You've got a bigger belly than you had before, and if you want- what do you want to get out of this?' And I said, 'I'd like to probably get my shape back, and just lose a little bit of weight…'' (Thomas)

### Theme 5: Autonomy: it has to be something I enjoy
Overall, men believed that an individual has to be personally motivated to make dietary and lifestyle changes so that they would be maintained long-term. Men who made changes to their diet and lifestyle out of necessity due to comorbidities such as diabetes, felt restricted and

thus sometimes broke the rules. Although the proposed dietary intervention which required men to increase intake of tomatoes was acceptable for most men, they expressed different preferences in regards to taste and ways of incorporating tomatoes in their diet. A few men favoured whole tomatoes over lycopene supplement (tablets) because they were suspicious of its content and production process.

'It would be easier, but- but, I mean, it's- yeah, it would be easier, without a doubt. If you could take a tomato tablet that does it, but then it's what goes into making the tablet. With a tomato, a fresh tomato, you know what you're getting, don't you?' (Thomas)

Similarly, men in general were amenable to a dairy-free diet but they had reservations about substituting dairy products with soya, because they were unfamiliar with soya products and perceived dairy substitutes as unnecessary due to a low dairy intake. Nevertheless, there were a few men who were hesitant to give up milk which they usually consumed with breakfast cereals or in porridge. Some men also commented that they have to enjoy or like the food to eat it, regardless of its health benefits. One man rejected the idea of eating tomatoes every day despite being told of the potential protective effect of tomatoes on prostate cancer:

'No I won't eat all that…normally I use one for the salad - if I make salad – [I wouldn't want to do]… You know? I don't eat every day…I don't feel like doing it, and I don't want to do it. (Laugh)' (Irvin)

Finally, the majority of the men interviewed preferred to walk on their own instead of taking part in a group exercise, even though they recognised the support and motivation that it could provide. Men explained that each person has his own schedule and routine, thus it is easier to walk on their own and at a pace that suited them.

'Na, because again I have to, they want, when I'm ready to do it they won't be ready, that's the problem. That's why I have to do my own.' (Albert)

### DISCUSSION
Most men in our study did not change their diet and lifestyle after a prostate cancer diagnosis. The lack of change was underpinned by five key themes: pre-cancer diet and lifestyle, evidence, coping with prostate cancer, ageing, and autonomy. The implications for research and practice are described below.

### Design of a dietary and lifestyle intervention for prostate cancer survivors
The proposed dietary and physical activity intervention was acceptable to men in this study, but they suggested it should be aimed at men with an elevated or rising PSA level, including those who are on active monitoring. The majority of dietary and lifestyle interventions for cancer

survivors have been conducted after treatment with the aim of improving general health and quality of life, but this finding suggests that men may be more interested in the clinical effect of interventions. Consequently, men perceived the proposed intervention as unnecessary because their PSA level was kept under control by the treatments they had received, which supports the findings of a previous study[28] and highlights the significance of using PSA as an outcome measure, despite criticism of its usefulness as a surrogate end-point in prostate cancer trials.[35 36] It also illustrates the challenges of informing men that dietary or lifestyle factors may act on prostate cancer independently of PSA.

This study also showed that men valued autonomy in their food and lifestyle choices and they might change their diet and lifestyle if it was deemed necessary and beneficial for health. Similar studies examining healthy eating using gender theory found that resistance among men was linked to masculine ideals of autonomy and rationality.[37] The authors suggest that healthy eating initiatives and materials could be tailored to men by emphasising personal choices and responsibility.[37] Most men in our study preferred individual over group exercise, which differed from previous studies reporting that men enjoyed the opportunity to socialise when exercising.[24 38] Nonetheless, men were willing to participate in weekly group exercise provided that a time and place that is convenient for all could be found. Therefore, either a choice of home-based intervention only or in combination with weekly supervised group exercise may appeal to African Caribbean prostate cancer survivors.

### Provision of dietary and lifestyle advice and support

Evidence from existing literature suggests that most men were uncertain about the role of diet and lifestyle in preventing prostate cancer progression or recurrence, but a strong belief in cancer–diet relationship was a predictor of dietary changes.[20–22 28 29] As with previous studies,[20 23 24 28 29] our findings suggest that advice from health professionals can strengthen men's beliefs on the relationship of prostate cancer with diet and lifestyle, influencing men to make positive lifestyle changes. Most men in our study did not recall receiving dietary or lifestyle information from health professionals, perhaps due to health professionals' hesitance on providing dietary and lifestyle advice as there is a lack of evidence on diet and prostate cancer and low awareness of relevant guidelines for cancer survivors.[28 39] Nonetheless, it is becoming clear that obesity is associated with higher risk of advanced and fatal prostate cancer,[11] and health professionals could potentially play a role in influencing men's health behaviour.

Awareness of supporting services was low among our participants. The UK National Cancer Survivorship Initiative recommend a 'recovery package' which consists of a combination of interventions, including a patient education and support event (health and wellbeing clinic) to be offered to all cancer survivors.[40] A report by the London Cancer Alliance found that there was no standardised definition or structure for health and wellbeing clinics in their partner NHS trust providers.[41] Therefore, more research is needed in designing and evaluating the referral pathways for healthy lifestyle services and developing a coherent model for health and wellbeing clinics.

### Active ageing and body weight

In line with previous research,[42 43] misconceptions about the harms or risks of strenuous physical activity in old age such as heart attack and feeling breathless were shared by men in this study. Older men also felt that going to the gym was incompatible with their age as for them it was typically associated with young people 'pumping iron'. Although there has been much effort in the UK and developed countries to promote active ageing, this shows that further work is needed to challenge prevailing social norms on ageing and physical activity to ensure men stay active as they age. Furthermore, older African Caribbean prostate cancer survivors should be informed that resistance exercise is beneficial to their health especially for preventing loss of muscle mass in men on androgen deprivation therapy.[14]

Ageing or being diagnosed with prostate cancer also heightened men's awareness of their health and prompted several men to make positive changes to their diet and lifestyle for maintenance of general health. Concerns about body weight among older men and prostate cancer survivors have been reported in several studies.[21 28 37] In this study, men wanted to maintain a healthy weight to avoid having a 'big belly'. This differs from previous studies which found that men lose weight for health purposes rather than for appearance (body image).[44 45] Men in this study watched their weight by eating smaller portion sizes, having light meals (less starchy food) and increasing their physical activity, contradicting previous studies which found that men preferred to lose weight through exercising because it is regarded as a masculine pursuit,[44] while 'dieting' practices such as increasing fruits and vegetable intake and reducing food portion sizes have been perceived to be feminine.[45 46]

Younger men also expressed interest in losing weight and regaining fitness after a period of inactivity due to side-effects of treatment. Similar to previous studies, men who experienced urinary incontinence were unsure about the type and intensity of physical activity that they could perform for fear of exacerbating their condition.[23 24 38] Thus, a supervised physical activity intervention would appeal to men to help them to lose weight and alleviate the side effects of treatments. A physical activity intervention could also be framed as a way for men to take charge of their recovery for better health.

Our study is one of the first to explore the dietary and lifestyle practices and information seeking behaviour of African Caribbean men after a prostate cancer diagnosis in the UK. Additionally, views and preferences for a dietary and physical activity intervention were sought from men to inform the design of a tailored intervention.

It included men from diverse sociodemographic backgrounds and with different prostate cancer clinical history.

We struggled to recruit men on active monitoring. It has been reported that Black African and Caribbean men in the UK were more likely to undergo radical treatment compared with their White counterparts,[47] perhaps due to a tendency for their prostate cancers to be managed more aggressively by clinicians.[48] Additionally, the proportion of men on active monitoring is relatively low—12% in UK in 2009[48]—compared with other prostate cancer treatments. We did not interview men's partners. It would be of interest to explore eating habits and food choices of the family with their partners to gain a deeper understanding of the roles and responsibilities regarding food within the household and contrast with the diet of single/widowed men who live alone. There was an indication that men preferred -consuming tomatoes over a lycopene supplement, which requires further investigation. Furthermore, questions on format of delivery (telephone-based, print-based, supervised physical activity) and ways of monitoring adherence (food diary, pedometers) were not explored in all interviews, precluding comparisons between men. In future research, clear and simple visual aids could be used to present and facilitate discussions on the proposed intervention. Finally, this study reflects only intentions in response to a dietary and physical activity intervention; this may differ from that in a real intervention.

## CONCLUSION

A prostate cancer diagnosis did not trigger dietary and lifestyle changes in this group of men who mostly perceived their diet and lifestyle to be healthy and were uncertain about the therapeutic benefits of diet and lifestyle on prostate cancer recurrence. They considered an intervention as unnecessary because their PSA level was kept under control by the treatments they had received. Men believed dietary and lifestyle changes should be self-initiated and motivated but were willing to make additional changes if they were perceived to be beneficial to health. Nonetheless, some men cited advice from health professionals and social support in coping with prostate cancer as facilitators to positive dietary and lifestyle changes. Furthermore, a prostate cancer diagnosis and ageing heightened men's awareness of their health, particularly in regards to their body weight. Therefore, a dietary and physical activity intervention which enhances men's autonomy, framed as helping men to regain fitness and aid post-treatment recovery and is aimed at men with elevated PSA may be appealing and acceptable to African Caribbean prostate cancer survivors.

**Author affiliations**
[1]School of Social and Community Medicine, University of Bristol, Bristol, UK
[2]NIHR Bristol Nutrition Biomedical Research Unit, University Hospitals Bristol Education & Research Centre, Bristol, UK
[3]MRC Integrative Epidemiology Unit, Bristol, UK
[4]Bristol Urological Institute, North Bristol NHS Trust, Southmead Hospital, Bristol, UK
[5]Department of Urology, Barts Health NHS Trust, The Royal London Hospital, London, UK
[6]School of Health Sciences, University of London, London, UK

**Acknowledgements** The authors would like to thank all participants for sharing their views and experience; Dr Mona Jeffreys and the Bristol Nutrition BRU prostate patient and public involvement group for their input into the study design and documents and all staff who had helped with the recruitment of participants, especially Dr Karen Tipples.

**Contributors** VE, ES, JAL and RM conceived the study. VE, ES, FC, RP and VN contributed to the design and conduct of the research and acquisition of data. VE and ES analysed the data. VE, ES, RM and JAL wrote the first draft of the paper. ES, JAL and RM provided supervision. VE, ES, JAL and RM have primarily responsibility for final content. All authors were involved in revising the paper and read and approved the final version of the paper.

**Funding** This work was supported by the National Institute for Health Research (NIHR) Bristol Nutrition Biomedical Research Unit. The unit is a partnership between University Hospitals Bristol NHS Foundation Trust and the University of Bristol. VE was a recipient of a 3-year PhD studentship at the Bristol Nutrition Biomedical Research Unit from 2012–2015. RM and JAL are recipients of CRUK programme funding (C18281/A19169).

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
