## [Reviewer comments · BMJ Open]

ARTICLE DETAILS

TITLE (PROVISIONAL)	Barriers and facilitators to healthy lifestyle and acceptability of a dietary and physical activity intervention among African Caribbean prostate cancer survivors in the UK: a qualitative study
AUTHORS	Er, Vanessa; Lane, Athene; Martin, Richard; Persad, Raj; Chinegwundoh, Frank; Njoku, Victoria; Sutton, Eileen

VERSION 1 – REVIEW

REVIEWER	Veronica Nanton University of Warwick United Kingdom
REVIEW RETURNED	06-May-2017

GENERAL COMMENTS	This is interesting and clearly written paper and the authors have done well to overcome the challenges to recruitment. Research into all aspects of prostate cancer on African Caribbean men in the UK is very much needed. As it stands, there are some significant limitations and some revisions, particularly to the background and the section on diet, are needed. Major The aim of the study is explained as an exploration of facilitators and barriers to a diet and exercise intervention and of the acceptability of such an intervention amongst African Caribbean men. The overarching limitation to this account of the study is the lack of attention to socio-cultural context. Apart from evidence of increased risk the paper unfortunately cites no other studies of African Caribbean men in relation to prostate cancer. There are not a large amount of UK studies but there are some (see Pedersen et al 2012). These should be cited and the gaps in understanding identified. There appears to be an a priori assumption that a particular dietary intervention is indicated and this is based on extrapolations from studies which, as the authors acknowledge, largely report American general prostate population data. The importance of understanding men's beliefs, behaviours and their context should be discussed in the introduction providing a rationale for the investigation. For an example from another condition see (Brown K, Avis M, Hubbard M. Health beliefs of African–Caribbean people with type 2 diabetes: a qualitative study. The British Journal of General Practice. 2007;57(539):461-469) The back ground section would be greatly improved by the inclusion of some reference to and discussion of traditional Caribbean diet and any evidence as to how this has been modified in the UK. Data on participants' diets (which is stated on p7 has been collected as
---

part of the research), possible changes over time and reasons for change would provide a valuable context for the discussion of the proposed dietary modification of additional lycopene and reduction in dairy products. As it stands the rationale for the proposed dietary intervention appears rather weak. The quote by Jonah p11 line 27 illustrates the point, 'well erm change to what.....'

Traditional Caribbean island diets have both benefits and negative aspects. Food is fresh and unprocessed and vegetables are plentiful. However the diet tends to be high in salt and in saturated fat and carbohydrate. Tomatoes are a common ingredient but little dairy produce is generally consumed. First generation migrants tend mainly to retain dietary habits from their country of origin. Some of the men interviewed have made small adjustments in response to their cancer and concerns over weight (9 participants were on hormone treatment and therefore likely to have gained weight) or in one instance in response to a co-morbidity but in general perceive their diets to be healthy. The evidence on protective effect of lycopene is strong while the association between dairy and prostate cancer has not been firmly established, thus the apparent scepticism of the men (and the ambivalence of their health care teams) is unsurprising. African Caribbean people suffer disproportionately from stroke associated with high salt consumption. The authors need to explain why the dietary intervention for this particular population group is focussed on lycopene and dairy products in relation to prostate cancer rather than on reinforcing general healthy eating (building on the positive aspects of men's diets) with an emphasis on the value of weight control in terms of cancer for which there is a substantial body of evidence. There is an acknowledgment in the conclusion that an intervention framed as improving fitness and aiding recovery is likely to be acceptable (however the question remains as to why this was not the starting point rather than a conclusion.

The authors acknowledge that it may have been valuable to interview wives but this is described as being useful to 'uncover discrepancies' (P4, L37). The purpose of interviewing partners would be better explained in terms of gaining a deeper understanding of beliefs around food and roles and responsibilities regarding food within the household. Understanding these complex issues is critical to any kind of dietary intervention. In addition the impact of living alone on diet among the widowers should also have been considered.

The authors state that information on socioeconomic positions (P6,L48) was sought but apart from one participant identified as a builder and another self employed nothing other than age or marital status is reported and there is only one mention of a comorbidity. Comorbidities are important with respect to their potential impact on diet (eg diabetes) and exercise and also with respect to men's priorities. A co-morbidity may at certain times be more salient than a prostate cancer which a man regards as cured or under control and this in turn may affect attitudes to lifestyle.

The discussion of exercise is interesting. Clearly offering support and encouragement for men to become more active or regain earlier levels of fitness is very important –though again some of the men were already incorporating the proposed intervention in their daily lives. Problems caused by continuing incontinence were very limiting for some of the men and for these men targeted exercises would certainly of value as well as the brisk walk. Incontinence and sexual function are clearly highly salient to men following prostatectomy so interventions aimed at these would very likely be welcomed.

The discussion (p17,L52) describes' the challenge of informing and

	convincing men that dietary or lifestyle factors may act on prostate cancer independently of PSA'. This sentence exemplifies a central problem of the paper- the authors have tried to convince men of the value of an intervention with insufficient evidence that the intervention- which may be beneficial in other terms will impact on the return or progress of their prostate cancer. The focus should be on encouraging men to build on positive aspects of their lifestyle which will help their weight management as weight is known to be important in carcinogenesis and to consume foods which maybe protective and will have general health benefits and limit those which do not. This would be an honest approach and would reflect the current state of knowledge. Minor P4 I 45 'Black British' please provide a definition The topic guide should be included How was the 'intervention' described to the men-what did the term mean to them The paper describes how men respond to the ideas of an intervention and explores some interesting ideas around autonomy and self- determination however the comment 'an intervention could be 'sold to men as killing two birds with one stone ' (p20 I27)is somewhat at odds with the concept of autonomy and should be rephrased. P11 L 39,L40 plurals 'are' and 'they ' needed P11 L 54 supporting quote needed P4 L23-28 Strengths and limitations A reference is needed for the statement that African Caribbean men have a preference for curative treatment . Please consider the use of terms such as 'convince' and 'persuade'. Focus on the need to provide clear straight forward information. Acknowledge the need to distinguish to patients between confirmed and tentative associations.
--	--

REVIEWER	Stephanie Bonn Karolinska Institutet, Sweden
REVIEW RETURNED	14-Jun-2017

GENERAL COMMENTS	This is a well written manuscript describing a quantitative study aiming to explore facilitators and barriers to changes in dietary behaviors and physical activity among African Caribbean prostate cancer survivors. The manuscript is interesting and reads well. Please find some specific comments listed below:  1. How did the change from purposive sampling to convenience sampling impact results and generalizability of results? 2. Since a lot of the results concern body weight it would be interesting to know for example the BMI of the men included in the study. Do the authors have information on BMI? 3. In the results, first paragraph, it would be helpful to the reader if the authors could summarize the characteristics presented in Table 1 briefly, for example include the mean age, years since diagnosis and BMI if that information is available. 4. A limitation to the study, that the authors also acknowledge, is that no men on active surveillance were included in the study. How large is the proportion of men on active surveillance in this population? 5. Discussion, first paragraph, please state the 5 key themes again instead of referring back to the text. 6. How many participants did the authors aim to include in the study? Are 14 participants sufficient to be able to draw conclusions?
---

VERSION 1 – AUTHOR RESPONSE

Reviewer: 1

Reviewer Name: Veronica Nanton

Institution and Country: University of Warwick, United Kingdom

Please state any competing interests: None declared

Please leave your comments for the authors below

This is interesting and clearly written paper and the authors have done well to overcome the challenges to recruitment. Research into all aspects of prostate cancer on African Caribbean men in the UK is very much needed.

As it stands, there are some significant limitations and some revisions, particularly to the background and the section on diet, are needed.

Major

The aim of the study is explained as an exploration of facilitators and barriers to a diet and exercise intervention and of the acceptability of such an intervention amongst African Caribbean men.

The overarching limitation to this account of the study is the lack of attention to socio-cultural context. Apart from evidence of increased risk the paper unfortunately cites no other studies of African Caribbean men in relation to prostate cancer. There are not a large amount of UK studies but there are some (see Pedersen et al 2012). These should be cited and the gaps in understanding identified.

R: We thank the reviewer for pointing this out. We have described Black African and Black Caribbean men's awareness and perceptions of prostate cancer, referencing the study by Pedersen and colleagues (P4 L25 to P5, L1-9).

There appears to be an a priori assumption that a particular dietary intervention is indicated and this is based on extrapolations from studies which, as the authors acknowledge, largely report American general prostate population data. The importance of understanding men's beliefs, behaviours and their context should be discussed in the introduction providing a rationale for the investigation. For an example from another condition see (Brown K, Avis M, Hubbard M. Health beliefs of African–Caribbean people with type 2 diabetes: a qualitative study. *The British Journal of General Practice*. 2007;57(539):461-469)

R: We have included a brief discussion on men's beliefs in the introduction as suggested by the reviewer (P5, L16-21).

The background section would be greatly improved by the inclusion of some reference to and discussion of traditional Caribbean diet and any evidence as to how this has been modified in the UK.

R: We thank the reviewer for the suggestions. We have included in the Background section further information on the diet of the African Caribbean population in the UK (P6, L17-22).

Data on participants' diets (which is stated on p7 has been collected as part of the research), possible changes over time and reasons for change would provide a valuable context for the discussion of the proposed dietary modification of additional lycopene and reduction in dairy products. As it stands the rationale for the proposed dietary intervention appears rather weak. The quote by Jonah p11 line 27 illustrates the point, 'well erm change to what.....'

R: We did not assess the diet of participants. Dietary intake/practices and rationale for changes to diet (if any) before and after a prostate cancer diagnosis were explored only in the interviews with men and are described in the manuscript. Since we did not have knowledge of participants' diet prior to the interview, we based the components of a proposed dietary intervention on the latest evidence from the literature (WCRF/AICR Report 2007) as outlined in the Introduction (P5 L23-25 to P6 L1-6).

Traditional Caribbean island diets have both benefits and negative aspects. Food is fresh and unprocessed and vegetables are plentiful. However the diet tends to be high in salt and in saturated fat and carbohydrate. Tomatoes are a common ingredient but little dairy produce is generally consumed. First generation migrants tend mainly to retain dietary habits from their country of origin. Some of the men interviewed have made small adjustments in response to their cancer and concerns over weight (9 participants were on hormone treatment and therefore likely to have gained weight) or in one instance in response to a co-morbidity but in general perceive their diets to be healthy. The evidence on protective effect of lycopene is strong while the association between dairy and prostate cancer has not been firmly established, thus the apparent scepticism of the men (and the ambivalence of their health care teams) is unsurprising. African Caribbean people suffer disproportionately from stroke associated with high salt consumption. The authors need to explain why the dietary intervention for this particular population group is focussed on lycopene and dairy products in relation to prostate cancer rather than on reinforcing general healthy eating (building on the positive aspects of men's diets) with an emphasis on the value of weight control in terms of cancer for which there is a substantial body of evidence. There is an acknowledgment in the conclusion that an intervention framed as improving fitness and aiding recovery is likely to be acceptable (however the question remains as to why this was not the starting point rather than a conclusion.

R: Although we were aware that a traditional Caribbean diet is rich in fresh fruits and vegetables, we did not want to make any assumptions about the diet of the participants and their views on the causes of prostate cancer (i.e. if it is diet-what are the food/nutrients linked to prostate cancer).

We based the proposed dietary intervention on:

a) WCRF/AICR 2007 report, which was the most up-to-date and authoritative evidence at the time of developing and conducting the study.

b) findings from previous studies (Horwood et al. 2014, Davison, B. J., & Breckon, E. (2012). Factors influencing treatment decision making and information preferences of prostate cancer patients on active surveillance. *Patient education and counseling*, 87(3), 369-374) show that prostate cancer survivors have a preference for prostate cancer-specific dietary advice/information. As a result, we decided on nutrients/foods linked to prostate cancer as the components of the proposed intervention. We agree with the reviewer about the importance of weight control for the health outcomes of cancer survivors. However, the evidence linking weight and prostate cancer was only established in late 2014 (WCRF/AICR Continuous Update Project), after the interviews had begun. We did not anticipate the concerns that men had about their body weight, these emerged during the early interviews so we amended the interview topic guide to incorporate these findings, as described in the Methods section (P8 L15-23).

NB: The interview period was incorrect and has been amended (P8 L4-5)

The authors acknowledge that it may have been valuable to interview wives but this is described as being useful to 'uncover discrepancies' (P4, L37). The purpose of interviewing partners would be better explained in terms of gaining a deeper understanding of beliefs around food and roles and responsibilities regarding food within the household.

Understanding these complex issues is critical to any kind of dietary intervention. In addition the impact of living alone on diet among the widowers should also have been considered.

R: We have removed 'uncover discrepancies', and made the changes as suggested by the reviewer (P4 L16-17, and P23 L1-3).

The authors state that information on socioeconomic positions (P6,L48) was sought but apart from one participant identified as a builder and another self employed nothing other than age or marital status is reported and there is only one mention of a comorbidity. Comorbidities are important with respect to their potential impact on diet (eg diabetes) and exercise and also with respect to men's priorities. A co-morbidity may at certain times be more salient than a prostate cancer which a man regards as cured or under control and this in turn may affect attitudes to lifestyle.

R: We have information on occupation but omitted this due to concerns of identification. Readers can

easily surmise participants' area of residence as the affiliation/institution of the health professionals who collaborated in the study is stated in the manuscript. Providing information on participants' occupation would make them more identifiable. However, we have now summarised the occupation of participants in a sentence in the Results section (P9 L18-23), and added a column on self-reported co-morbidities in Table 1 (P10).

The discussion of exercise is interesting. Clearly offering support and encouragement for men to become more active or regain earlier levels of fitness is very important –though again some of the men were already incorporating the proposed intervention in their daily lives. Problems caused by continuing incontinence were very limiting for some of the men and for these men targeted exercises would certainly be of value as well as the brisk walk. Incontinence and sexual function are clearly highly salient to men following prostatectomy so interventions aimed at these would very likely be welcomed.
R: We agree with the reviewer.

The discussion (p17,L52) describes 'the challenge of informing and convincing men that dietary or lifestyle factors may act on prostate cancer independently of PSA'. This sentence exemplifies a central problem of the paper- the authors have tried to convince men of the value of an intervention with insufficient evidence that the intervention- which may be beneficial in other terms will impact on the return or progress of their prostate cancer. The focus should be on encouraging men to build on positive aspects of their lifestyle which will help their weight management as weight is known to be important in carcinogenesis and to consume foods which maybe protective and will have general health benefits and limit those which do not. This would be an honest approach and would reflect the current state of knowledge.

R: The concerns that men had about their body weight and image emerged during the early interviews, before there was strong evidence linking body weight and prostate cancer progression (as stated above). However, we amended the topic guide during the study to reflect this. As noted above, drawing from available evidence, our proposed intervention is based on encouraging men to consume foods that maybe protective (foods rich in lycopene, e.g. tomatoes) and limit those that may not be (dairy).

Minor

P4 | 45 'Black British' please provide a definition

R: We have provided the definition for Black men in brackets and updated the statistics and reference for prostate cancer incidence and mortality in England only (see P4, L25). The classification for ethnicity is based on Census and Hospital Episode Statistics.

The topic guide should be included

R: We have now included a topic guide.

How was the 'intervention' described to the men-what did the term mean to them

R: We were conscious that the word 'intervention' is used widely in the medical field but not in day-to-day conversations, so the interviewer (VE) did not use it in the interview. VE asked men about the intervention by framing her question as "If you were asked to eat more tomatoes (cut out dairy/walk briskly for 30 minutes a day) ..." or "We want to find out if tomato has an impact on prostate cancer, would you eat more tomatoes..."

The paper describes how men respond to the ideas of an intervention and explores some interesting ideas around autonomy and self- determination however the comment 'an intervention could be 'sold to men as killing two birds with one stone' (p20 |27) is somewhat at odds with the concept of autonomy and should be rephrased.

R: We thank the reviewer for highlighting this. We have removed the phrase 'sold to men as killing

two birds with one stone' (P22, L9)

P11 L 39,L40 plurals 'are' and 'they ' needed

R: We have made these changes.

P11 L 54 supporting quote needed

R: We have added a supporting quote (P13 L17-20). The quote was included in the table in Supplementary Material previously.

P4 L23-28 Strengths and limitations A reference is needed for the statement that African Caribbean men have a preference for curative treatment.

R: We have rephrased the sentence as 'Black men were more likely to undergo radical treatment' as reported by Shlomo and colleagues (P4 L10-13 and P22 L20-24).

Please consider the use of terms such as 'convince' and 'persuade'. Focus on the need to provide clear straight forward information. Acknowledge the need to distinguish to patients between confirmed and tentative associations.

R: We have removed the terms 'convince' and 'persuade'. It should be noted that there is consistent and strong evidence that resistance exercise prevents muscle mass loss among men on androgen-deprivation therapy (Gardner, Jason R., Patricia M. Livingston, and Steve F. Fraser. "Effects of exercise on treatment-related adverse effects for patients with prostate cancer receiving androgen-deprivation therapy: a systematic review." *Journal of Clinical Oncology* 32.4 (2013): 335-346).

Reviewer: 2

Reviewer Name: Stephanie Bonn

Institution and Country: Karolinska Institutet, Sweden

Please state any competing interests: None declared

Please leave your comments for the authors below

This is a well written manuscript describing a quantitative study aiming to explore facilitators and barriers to changes in dietary behaviors and physical activity among African Caribbean prostate cancer survivors. The manuscript is interesting and reads well. Please find some specific comments listed below:

1. How did the change from purposive sampling to convenience sampling impact results and cy of results?

R: The changes from purposive sampling to convenience sampling would have resulted in a narrower sample, and that was reflected in the limitations of the study. However, the study findings and conclusions are still valid for the men we had spoken to.

2. Since a lot of the results concern body weight it would be interesting to know for example the BMI of the men included in the study. Do the authors have information on BMI?

R: No, we do not have information on BMI as this was a standalone qualitative study and we did not record anthropometric measurements.

3. In the results, first paragraph, it would be helpful to the reader if the authors could summarize the characteristics presented in Table 1 briefly, for example include the mean age, years since diagnosis and BMI if that information is available.

R: We thank the reviewer for the suggestion. We have summarised the characteristics of participants in the first paragraph of the Results section (P9 L18-23).

4. A limitation to the study, that the authors also acknowledge, is that no men on active surveillance were included in the study. How large is the proportion of men on active surveillance in this population?

R: A report on treatment routes of prostate cancer showed that the proportion of Black men in UK on active surveillance was 12% in 2009 (Hounscome et al 2012). We have added this information in the limitations (P22 L23-24).

5. Discussion, first paragraph, please state the 5 key themes again instead of referring back to the text.

R: We have added the 5 key themes in the Discussion section (P19 L6-7).

6. How many participants did the authors aim to include in the study? Are 14 participants sufficient to be able to draw conclusions?

R: We aimed to recruit between 15 and 20 participants. The sample size was expected to obtain sufficient data to detect possible themes based on previous research and advice from experienced qualitative researchers. Nonetheless, no new theme was identified from the data after we had interviewed 14 participants. In other words, we have reached data saturation-the most common guiding principle for assessing the adequacy of sample-by 14 interviews.

The purpose of qualitative research is to gain an understanding of a phenomenon, is focused on the way people interpret and make sense of the world in which they live in, and is concerned with the quality and richness of the data. However, several studies (Henning et al 2017, Guest 2006) and a report of expert's opinions (Baker et al 2012) found that data saturation is reached by 9, 12 and 14 interviews respectively. Again, highlighting the difficulty of 'prescribing' a sample size for qualitative research.

Hennink et al. 2017. "Code saturation versus meaning saturation: How many interviews are enough?." Qualitative health research 27(4): 591-608

Guest et al. 2006. "How many interviews are enough? An experiment with data saturation and variability." Field methods 18(1): 59-82.

Baker et al. 2012. "How many qualitative interviews is enough?: Expert voices and early career reflections on sampling and cases in qualitative research."

VERSION 2 – REVIEW

REVIEWER	Veronica Nanton University of Warwick United Kingdom
REVIEW RETURNED	02-Aug-2017

GENERAL COMMENTS	The authors have greatly improved this paper having paid attention to the reviewers comments on the first version. It reads well and makes a valuable contribution. I have no further substantive suggestions. I have two very minor observations; There is a letter or a word missing on p6 l17 (African Caribbean). Please give the age ranges of the participants and 52-80 and a median rather than a mean. I hope the authors will consider following up this qualitative investigation with an intervention study.
---

VERSION 2 – AUTHOR RESPONSE

Reviewer's Comments

1. There is a letter or a word missing on p6 l17 (African Caribbean).

R: We thank the reviewer for pointing this out. We have made the changes (P6, L19).

2. Please give the age ranges of the participants and 52-80 and a median rather than a mean.

R: We have provided the age ranges of the participants and the median age (P9, L18)